# Dual Defense: Enhancing Privacy and Mitigating Poisoning Attacks in Federated Learning

**Runhua Xu**
Beihang University
runhua@buaa.edu.cn

**Shiqi Gao**
Beihang University
gaoshiqi@buaa.edu.cn

**Chao Li**
Beijing Jiaotong University
li.chao@bjtu.edu.cn

**James Joshi**
University of Pittsburgh
jjoshi@pitt.edu

**Jianxin Li**[*]
Beihang University and Zhongguancun Laboratory
lijx@buaa.edu.cn

## Abstract

Federated learning (FL) is inherently susceptible to privacy breaches and poisoning attacks. To tackle these challenges, researchers have separately devised secure aggregation mechanisms to protect data privacy and robust aggregation methods that withstand poisoning attacks. However, simultaneously addressing both concerns is challenging; secure aggregation facilitates poisoning attacks as most anomaly detection techniques require access to unencrypted local model updates, which are obscured by secure aggregation. Few recent efforts to simultaneously tackle both challenges offen depend on impractical assumption of non-colluding two-server setups that disrupt FL's topology, or three-party computation which introduces scalability issues, complicating deployment and application. To overcome this dilemma, this paper introduce a **D**ual **D**efense **Fed**erated learning (*DDFed*) framework. *DDFed* simultaneously boosts privacy protection and mitigates poisoning attacks, without introducing new participant roles or disrupting the existing FL topology. *DDFed* initially leverages cutting-edge fully homomorphic encryption (FHE) to securely aggregate model updates, without the impractical requirement for non-colluding two-server setups and ensures strong privacy protection. Additionally, we proposes a unique two-phase anomaly detection mechanism for encrypted model updates, featuring secure similarity computation and feedback-driven collaborative selection, with additional measures to prevent potential privacy breaches from Byzantine clients incorporated into the detection process. We conducted extensive experiments on various model poisoning attacks and FL scenarios, including both cross-device and cross-silo FL. Experiments on publicly available datasets demonstrate that *DDFed* successfully protects model privacy and effectively defends against model poisoning threats.

## 1  Introduction

Federated learning (FL)[18] is gaining popularity as a collaborative model training paradigm that provides primary privacy protection by eliminating the need of sharing private training data. Based on the participants' scale, FL is typically divided into two categories: cross-silo FL and cross-device FL[17]. Cross-device FL typically involves numerous similar devices, while cross-silo FL usually includes fewer participants like organizations. Recent studies show that FL mainly confronts two types of threats: privacy risks from curious adversaries attempting to compromise data privacy

---

[*]corresponding author

38th Conference on Neural Information Processing Systems (NeurIPS 2024).

through methods like membership inference and model inversion attacks, and security risks from Byzantine adversaries looking to damage the final model's integrity with backdoors or by lowering its accuracy [2, 24, 14, 11, 3, 1, 34].

To mitigate privacy risks in FL, researchers have developed a range of techniques to bolster privacy. These encompass differential privacy-based aggregation [32], as well as secure aggregation approaches using homomorphic encryption[41], functional encryption[35], and secure multi-party computation[6, 43]. Aside from privacy concerns, many studies have proposed strategies to identify and mitigate potentially harmful updates during the model aggregation phase, thereby safeguarding the global model against adversarial attacks. Notable Byzantine-resistant aggregation mechanisms encompass the Krum fusion method[5], cosine defense aggregation mechanism[29, 38], and median/mean-based strategies like clipping median and trimmed mean strategies [40]. Research in these two areas has been conducted separately, and addressing both issues at once continues to be challenging. This difficulty arises because secure aggregation makes it easier for adversarial attacks to occur, as most anomaly detection methods need access to "unencrypted" local model updates that secure aggregation protects.

Few recent efforts [39, 13, 16, 43, 15, 23, 9, 20] to tackle both challenges simultaneously often depend on differential privacy techniques [39, 13, 16, 22, 12], which can degrade model performance due to added noise, or rely on impractical non-colluded two-server assumption that disrupts FL's topology[43, 15, 23, 9, 20], complicating its deployment and application. In light of these limitations, a critical yet overlooked question is *how to create a straightforward dual defense strategy that simultaneously strengthens privacy protection and mitigates poisoning attacks without introducing new participant roles or altering the single-server multiple-clients structure?*

To address this dilemma, this paper proposes a **D**ual **D**efense approach that simultaneously enhances privacy protection and combats poisoning attacks in **Fed**erated learning (*DDFed*), without changing the structure of current FL frameworks. *DDFed* initially leverages cutting-edge cryptographic technology, specifically fully homomorphic encryption (FHE), to securely aggregate model updates without the impractical assumption of non-colluding two-server setups and ensures strong privacy protection by permitting only the aggregation server to perform secure aggregation in the dark. To tackle the challenge of detecting malicious models within encrypted model updates, *DDFed* introduces a novel two-phase anomaly detection mechanism. This approach enables cosine similarity computation over encrypted models and incorporates a feedback-driven collaborative selection process, with additional measures to prevent potential privacy breaches from Byzantine clients incorporated into the detection mechanism. Our main contributions are summarized as follows:

- We introduce a dual defense strategy that simultaneously boosts privacy and combats poisoning attacks in federated learning. This is achieved by integrating FHE-based secure aggregation with a mechanism for detecting malicious encrypted models based on similarity.

- To effectively detect malicious models in encrypted updates, we propose a novel two-phase anomaly detection mechanism with extra safeguards against potential privacy breaches by Byzantine clients during the detection process. Additionally, we introduce a clipping technique to bolster defenses against diverse poisoning attacks.

- We carried out comprehensive experiments on multiple model poisoning attacks and federated learning scenarios, covering both cross-device FL and cross-silo FL. Our experiments with publicly accessible datasets demonstrate *DDFed*'s effectiveness in safeguarding model privacy and robustly defending against model poisoning threats.

## 2   Related Works

**Privacy Risks and Countermeasures in FL**   The fundamental design of FL ensures that all training data stays with its owner, offering basic privacy. However, it still exposes vulnerabilities to inference attacks, which allow adversaries to extract information about the training data used by each party [24, 27, 2, 24, 14, 11]. In some cases, the risk of private information leakage may be unacceptable. Therefore, several defenses have been suggested to mitigate these risks, including differential privacy (DP) and secure aggregation (SA), based on various cryptographic primitives such as (partial) homomorphic encryption [21, 41], threshold Paillier [30], functional encryption [36], and pairwise masking protocols [6].

**Poisoning Risks and Countermeasures in FL.** Besides privacy inference attacks, FL is also susceptible to poisoning attacks, where adversaries can compromise certain clients and manipulate their data or models to intentionally worsen the global model's performance by introducing corrupted updates during training. This paper focuses on untargeted model attacks, whose goal is to significantly diminish the effectiveness of the global model through methods such as Inner Product Manipulation (IPM) attack [34], scaling attack[1], and "a little is enough" (ALIE) attack [3]. Several strategies have been developed to counteract the impact of attacks, ensuring they don't compromise model performance. These strategies fall into two categories: client-side and server-side defenses. Client-side defenses adjust the local training algorithm with a focus on secure client updates[28], whereas server-side defenses [5, 29, 38, 40] either reduce the influence of updates from malicious clients through adjusted aggregation weights or use clustering techniques to aggregate updates from trustworthy clients only. However, these defense strategies typically operate under the assumption that model updates are not encrypted, which contradicts the objectives of privacy-focused secure aggregation defense strategies.

**Private and Robust Federated Learning.** In privacy-preserving FL, identifying poisoning attacks is harder because of the need to balance local model privacy with the detection of harmful models. Only a few existing studies like those mentioned in [39, 13, 16, 15] employ Byzantine-resilient aggregation through differential-privacy techniques. This approach necessitates a compromise between privacy and model accuracy. Additionally, recent initiatives have been launched to address this problem through diverse methods by using various secure computation technologies. These include 3PC[9], which faces scalability limitations; an oblivious random grouping method constrained by its design for partial parameter disclosure[43]; and both additive secret sharing[20] and two-trapdoor homomorphic encryption[23], which depend on the impractical assumption of non-colluding dual servers.

## 3 Dual Defense Federated Learning Framework

### 3.1 Formulation and Assumption

**Formulation.** A typical FL framework involves $m$ clients, $\mathcal{C}_1, ..., \mathcal{C}_m$, and a single aggregation server $\mathcal{A}$. Each client $\mathcal{C}_i$ possesses its own dataset $D_i$. The overarching goal in FL across these $m$ clients is to minimize the global objective function:

$$\min_{\boldsymbol{W}_1, ..., \boldsymbol{W}_m} \frac{1}{m} \sum_{i=1}^{m} \frac{|D_i|}{\sum_{i=1}^{m} |D_i|} L_i(\boldsymbol{W}_i; D_i). \tag{1}$$

Here, $L_i$ represents the local loss function for each client's data, and $\boldsymbol{W}_i$ are the local model parameters specific to client $\mathcal{C}_i$. The term $D_i$ refers to the private dataset of client $i$, with $|D_i|$ indicating its size in terms of sample count. In short, the goal of general FL is to learn an optimal global model $\boldsymbol{W}_G$ across $m$ clients. This is achieved by periodically synchronizing the model parameters from all clients using specified fusion algorithms like *FedAvg* and its variants, with the aggregation server $\mathcal{A}$ over several training rounds.

Due to various malicious activities, including inference attacks that aim to steal private information from legitimate clients and poisoning attacks designed to undermine model integrity by degrading its performance, existing privacy-preserving FL often relies on a secure aggregation mechanism[21, 41, 30, 36, 6]. Typically, without loss of generality, during the $t$-th federated learning training round, each client $\mathcal{C}_i$ secures its local model update $\boldsymbol{W}_i$ - referred to as $[\![\boldsymbol{W}_i]\!]$ throughout this paper - before transmitting it to the aggregation server. This is achieved by using various privacy-enhancing technologies such as homomorphic encryption and secure multi-party computation.

**Threat Assumption.** *DDFed* tolerates an adversary, capable of corrupting any subset of local clients at a specified ratio $r_{\text{ATTACK}}, s.t., r_{\text{ATTACK}} < 0.5$, to carry out model poisoning attacks that degrade the global model's performance. Additionally, we assume the aggregation server $\mathcal{A}$ is semi-honest (honest-but-curious), meaning it adheres to the protocol but seeks to glean as much private information as possible. Similarly, the compromised clients $\mathcal{C}_i^{\text{ADV}}$ can conduct privacy inference attacks like those performed by $\mathcal{A}$. In summary, regarding privacy preservation, both the inquisitive $\mathcal{A}$ and the corrupted client subset aim to extract private information from benign clients; however,

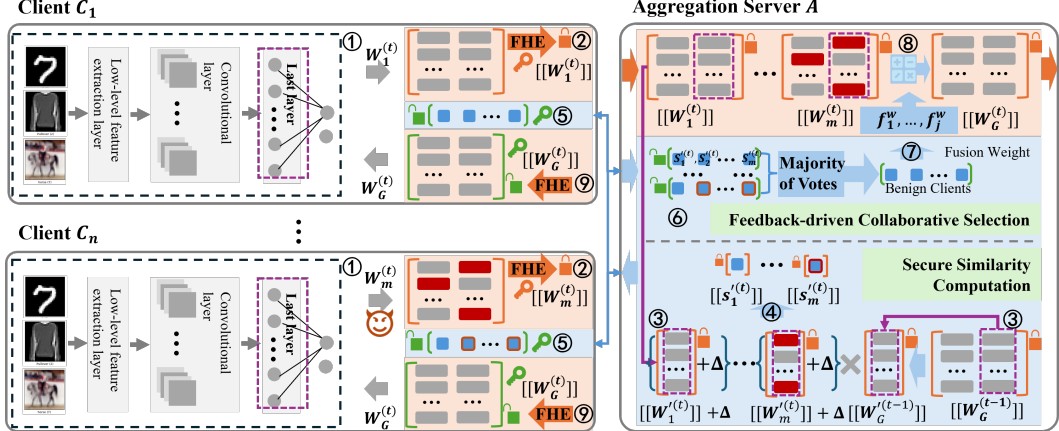

① Local Training ② Local Model Encryption ③ Last Layer Extraction ④ Secure Similarity Computation ⑤ Similarity Score Decryption
⑥ Feedback-driven Collaborative Selection ⑦ Fusion Weight Generation ⑧ FHE-based Secure Aggregation ⑨ Global Model Decryption

Figure 1: Overview of *DDFed* framework and illustration of a single round *DDFed* training.

only the corrupted client subset will also initiate model poisoning attacks to undermine the global model.

## 3.2 Framework Details

**Objective of DDFed.** *DDFed* is designed to bolster privacy protection and mitigate model poisoning attacks seamlessly within the existing FL framework. Unlike existing private and robust approaches [39, 13, 16, 43, 15, 23, 9, 20] that add new participant roles or depend on differential privacy, which may compromise model performance, *DDFed* maintains effectiveness efficiently. *DDFed* introduces a dual defense strategy that combines fully homomorphic encryption (FHE) for secure data aggregation with an optimized similarity-based mechanism to detect malicious models, ensuring unparalleled privacy protection and security against model poisoning attacks.

Similarity-based methods are commonly used in existing studies for anomaly detection models [29, 38]. Specifically, it computes the cosine similarity between each local model update of training round $t$ and the global model from the previous round $t-1$:

$$cos(\alpha_i) = \frac{\langle \boldsymbol{W}_i^{(t)}, \boldsymbol{W}_G^{(t-1)} \rangle}{\|\boldsymbol{W}_i^{(t)}\|_2 \cdot \|\boldsymbol{W}_G^{(t-1)}\|_2}, \tag{2}$$

where $\alpha_i$ denotes the angle between global model weights $\boldsymbol{W}_G^{(t-1)}$ and local model update $\boldsymbol{W}_i^{(t)}$ of client $\mathcal{C}_i$. However, existing similarity-based mechanisms [29, 38] offer no privacy protection for local model updates, and integrating FHE into them poses significant challenges. These challenges arise from FHE's limitations in performing division and comparison operations, which are essential for identifying benign clients in these methods.

**Framework Overview and Training Process.** Figure 1 provides an overview of *DDFed* framework, which includes several clients $\mathcal{C}_1, ..., \mathcal{C}_m$ and a single aggregation server $\mathcal{A}$, consistent with the architecture of most existing FL frameworks. In the following section, we demonstrate the *DDFed* training process. Due to space limitations, the formal algorithm pseudocode is provided solely in Appendix A.1.

Before the FL training begins, each client $\mathcal{C}_i$ is equipped with an FHE key pair (PK, SK). During the FL training phase, let's assume that in the $t$-th round, each client $\mathcal{C}_i$ trains a local model $\boldsymbol{W}_i^{(t)}$ (①) and performs the normalization and encryption as $[\![\boldsymbol{W}_i^{(t)}]\!] = \text{FHE.ENC}_{\text{PK}}(\boldsymbol{W}_i^{(t)})$ with public key PK (②). Upon receiving encrypted local models, $\{[\![\boldsymbol{W}_i^{(t)}]\!]\}_{i \in [1,...,m]}$, $\mathcal{A}$ starts to detect anomaly model updates over all encrypted local models. Specifically, $\mathcal{A}$ first extracts the last layer, denoted as $\{[\![\boldsymbol{W}_i^{'(t)}]\!]\}_{i \in [1,...,m]}$, which remains encrypted (③), and adds a perturbation $\Delta^{(t)}$ to safeguard against

potential privacy attacks by malicious clients. Next, it retrieves the last layer of the encrypted global model from the previous training round ($[\![\boldsymbol{W}_G^{'(t-1)}]\!]$), The method for adding perturbations will be discussed in Section 3.2. Then, $\mathcal{A}$ performs secure inner-product between each perturbed $[\![\boldsymbol{W}_i^{'(t)}]\!] + \Delta^{(t)}$ and $[\![\boldsymbol{W}_G^{'(t-1)}]\!]$ to derive encrypted similarity score, denoted as $[\![\boldsymbol{s}^{'(t)}]\!] = ([\![s_1^{'(t)}]\!], ..., [\![s_m^{'(t)}]\!])$, and query each client (④). After receiving $[\![\boldsymbol{s}^{'(t)}]\!]$, each client $\mathcal{C}_i$ decrypts it to obtain the plaintext scores $\boldsymbol{s}_i^{'(t)}$. Subsequently, each client submits their list of similarity scores (⑤). It's important to note that at this stage, malicious clients may tamper with their similarity scores in an attempt to prevent detection of their compromised models. Since a benign client will honestly and accurately decrypt and select trustworthy clients group via threshold-based filter, and hence their results should be consistent. Therefore, $\mathcal{A}$ uses a majority voting strategy to acquire the final client score list, i.e., the voted $\boldsymbol{s}^{(t)}$ (⑥). Next, $\mathcal{A}$ normalizes $\boldsymbol{s}^{(t)}$ and generates the fusion weight (⑦). Here, *DDFed* employs *FedAvg*'s approach by weighting the aggregation according to dataset size proportions in current training round (⑧). Finally, each client $\mathcal{C}_i$ receives the aggregated global model $[\![\boldsymbol{W}_G^{(t)}]\!]$, decrypts it, and initiates the $(t+1)$-th round of *DDFed* training (⑨).

**Private and Robust Malicious Model Detection.** As observed in [38], the distribution of local data labels can be more effectively represented in the weights of the last layer than in other layers. Consequently, *DDFed* employs a similar approach to enhance the efficiency of detecting anomalies, as it requires performing similarity computation on encrypted model updates. Given that FHE supports only basic mathematical operations, and the similarity-based anomaly model detection mechanism needs complex operations like division (as shown in equation 2), comparison and sorting operations, *DDFed* breaks it down into two stages: *secure similarity computation* and *feedback-driven collaborative selection*. In the rest of the paper and during our experimental evaluation, we adhere to the layer section settings described in [38]. However, *DDFed* can be easily extended to support strategies for detecting malicious models using full layers. Additional experiments are detailed in Appendix A.2.4 to demonstrate the impact of layer sections on the *DDFed* framework.

*Secure Similarity Computation.* To circumvent division operations, *DDFed* necessitates that all clients pre-process their inputs for normalization and shifts the task of comparing similarity scores to the client side. This is because clients possess the FHE private key, allowing them to obtain the similarity score in plaintext. Formally, we have the following:

$$[\![cos(\alpha_i)]\!] = \frac{\langle [\![\boldsymbol{W}_i^{(t)}]\!], [\![\boldsymbol{W}_G^{(t-1)}]\!]\rangle}{\|[\![\boldsymbol{W}_i^{(t)}]\!]\|_2 \cdot \|[\![\boldsymbol{W}_G^{(t-1)}]\!]\|_2} = \langle [\![\frac{\boldsymbol{W}_i^{(t)}}{\|\boldsymbol{W}_i^{(t)}\|_2}]\!], [\![\frac{\boldsymbol{W}_G^{(t-1)}}{\|\boldsymbol{W}_G^{(t-1)}\|_2}]\!]\rangle, \tag{3}$$

where each client $\mathcal{C}_i$ prepares the $\frac{\boldsymbol{W}_i^{(t)}}{\|\boldsymbol{W}_i^{(t)}\|_2}$ and $\frac{\boldsymbol{W}_G^{(t-1)}}{\|\boldsymbol{W}_G^{(t-1)}\|_2}$ in advance, and then encrypts them using FHE encryption algorithm. Next, the aggregation server $\mathcal{S}$ verifies received $[\![\frac{\boldsymbol{W}_G^{(t-1)}}{\|\boldsymbol{W}_G^{(t-1)}\|_2}]\!]$ and perturbs local inputs and conducts secure inner-product computation as follows:

$$[\![\boldsymbol{s}^{'(t)}]\!] = \langle [\![\frac{\boldsymbol{W}_i^{(t)}}{\|\boldsymbol{W}_i^{(t)}\|_2}]\!] + \Delta^{(t)}, [\![\frac{\boldsymbol{W}_G^{(t-1)}}{\|\boldsymbol{W}_G^{(t-1)}\|_2}]\!]\rangle. \tag{4}$$

*Motivation of Similarity Score Perturbation.* *DDFed* aims to simultaneously address privacy and poisoning risks. This means it not only considers model poisoning attacks but also prevents adversarial clients from inferring private information from other benign clients by exploiting decrypted similarity scores and previous global models. To mitigate this privacy risk, *DDFed* improves secure inner-product computation by introducing perturbations into each normalized and encrypted model update. Specifically, *DDFed* uses $(\varepsilon, \delta)$-differential privacy with a Gaussian mechanism as its method of perturbation, $\Delta^{(t)} = \mathcal{N}(0, \sigma^2), \sigma = \frac{\Delta f \sqrt{2\ln(1.25/\delta)}}{\varepsilon}$, where $(\varepsilon, \delta)$ represents the parameters of the DP mechanism and $\Delta f$ denotes sensitivity.

It's important to note that our perturbation affects only the anomaly detection phase and does not change the encrypted model updates that are to be aggregated. Consequently, the final aggregated model retains its accuracy, just as it would with a standard aggregation mechanism. Furthermore, our experiments indicate that the perturbation noise does not affect the effectiveness of anomaly detection.

Even at $\varepsilon = 0.01$, which offers strong privacy protection, *DDFed* still performs well and delivers good model performance.

***Feedback-driven Collaborative Selection.*** As shown in the threat model, *DDFed* tolerates less than 50% malicious clients, indicating that over half of the clients are benign and will execute the steps honestly and correctly as designed. *DDFed* employs a feedback-driven collaborative selection approach to filter out potentially malicious models. Specifically, upon receiving the encrypted $[\![\boldsymbol{s}'^{(t)}]\!]$, each client $\mathcal{C}_i$ first decrypts to acquire $\boldsymbol{s}_i'^{(t)}$ using the FHE private key SK. Next, each client $\mathcal{C}_i$ independently decrypts the similarity scores, sorts them, and selects trustworthy clients $\boldsymbol{s}_i^{(t)}$ for the current training round based on a threshold. *DDFed* uses only the mean value of similarity scores as its filtering threshold. Subsequent experiments have demonstrated its effectiveness. Additionally, *DDFed* is open and compatible with alternative methods for setting thresholds. After each client returns their decision on the group of benign clients ($\boldsymbol{s}_i^{(t)}$), the aggregation server uses a majority of vote strategy to decide the final aggregation group ($\boldsymbol{s}^{(t)}$) for the current training round. Next, similar to *FedAvg*, *DDFed* applies a data size-based fusion weight strategy to calculate each client's fusion weight $\boldsymbol{f}_{\boldsymbol{s}^{(t)}}^{\boldsymbol{W}^{(t)}}$ in the aggregation group, where $f_j^{(t)} = \frac{|D_j|}{\sum_{j \in \boldsymbol{s}^{(t)}} |D_j|}$.

**FHE-based Secure Aggregation with Clipping.** *DDFed*'s secure aggregation leverages the FHE cryptosystem, specifically the CKKS instance[8], which excels in arithmetic operations on encrypted real or complex numbers and stands as one of the most efficient methods for computing with encrypted data. Formally, the aggregation server performs secure aggregation as $[\![\boldsymbol{W}_G^{(t)}]\!] = \langle [\![\boldsymbol{W}^{(t)}]\!], \boldsymbol{f}_{\boldsymbol{s}^{(t)}}^{\boldsymbol{W}^{(t)}} \rangle$. Once receiving the aggregated global model $[\![\boldsymbol{W}_G^{(t)}]\!]$, each client $\mathcal{C}_i$ uses their private key to decrypt it, obtaining the final global model $\boldsymbol{W}_G^{(t)}$ in plaintext via the FHE decryption algorithm. In contrast to current approaches in private and robust FL, *DDFed* uniquely enables each benign client to execute a clipping operation before the next training round. This enhancement is designed to counteract more sophisticated model poisoning attacks that conventional similarity-based methods [29, 38] fail to address, as will be shown in the experiments section.

### 3.3 Analysis on Privacy and Robustness

Based on the threat model discussed earlier, *DDFed* prevents an honest-but-curious aggregation server from potentially inferring private information from accessible model updates. Additionally, it also withstands a subset of local clients, compromised by an adversary, to launch model poisoning attacks and attempt to infer private information from other benign clients during the anomaly model detection phase.

In terms of privacy risks, *DDFed* utilizes FHE primitives to ensure cryptographic-level privacy protection. This means the aggregation server processes each operation without any insight into the model update (in the dark), eliminating any chance of inferring private information from local model updates. Furthermore, to counter potential inferences by corrupted clients exploiting decrypted similarity scores, *DDFed* incorporates a perturbation method where DP noise is added during the secure similarity computation phase. Due to space limitations, the formal DP-enhanced perturbation analysis is provided solely in Appendix A.3.

Regarding the risk of poisoning attacks, *DDFed* adopts similarity-based anomaly detection technologies with additional optimizations such as perturbation-based similarity computation and post-aggregation clipping. These enhancements bolster the robustness of its aggregation mechanism. Our experiments demonstrate that *DDFed* effectively resists a range of continuous poisoning attacks, including IPM, SCALING, and ALIE attacks, which will be elaborated in Section 4.

## 4 Experiments

### 4.1 Experimental Setup

**Datasets and Implementation.** We assessed our proposed *DDFed* framework using publicly available benchmark datasets: MNIST[19], a collection of handwritten digits, and Fashion-MNIST (FMNIST)[33], which includes images of various clothing items, offering a more challenging and

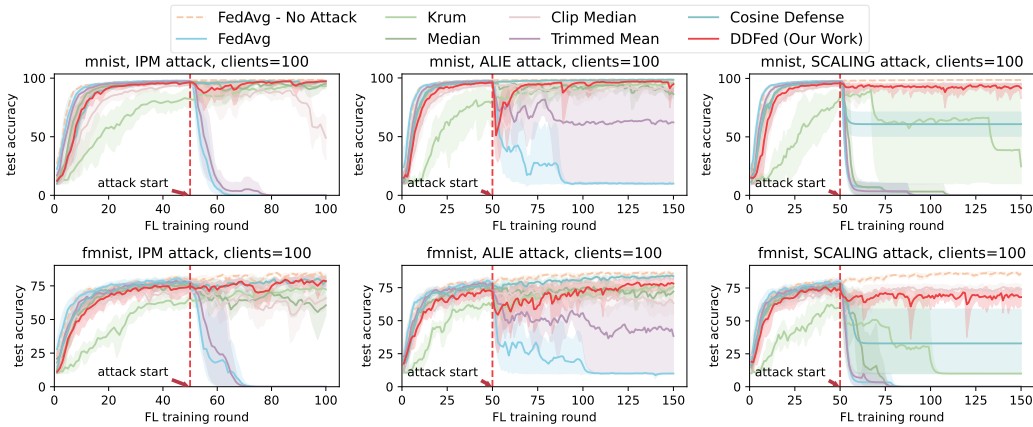

Figure 2: Comparison of defense effectiveness across various defense approaches, evaluated on MNIST (top) and FMNIST(bottom), under IPM attack (left), ALIE attack (middle), and SCALING attack (right).

diverse dataset for federated learning tasks. We create non-iid partitions for all datasets based on previous research [38, 43], using a default $q$ value of 0.5, where a higher $q$ reflects greater degrees of non-iid. We assess the framework's performance using a nine-layer CNN model with 225k parameters, secured by the FHE cryptosystem in each training round. This secure aggregation is implemented through TenSEAL library [4]. The experimental *DDFed* is available on the GitHub repository.

**Baselines and Default Setting.** We compare our proposed method ***DDFed*** with well-known FL fusion algorithms and robust aggregation methods, including ***Krum*** [5], ***Cos Defense*** [38], and median/mean-based approaches like ***Median***, ***Clipping Median***, and ***Trimmed Mean*** strategies[40]. We exclude baselines such as FLTrust[7] or RFFL[37] because they require server-side validation data or are incompatible with client sampling, making them impractical for real-world applications. Additionally, we omit secure robust approaches[9, 20, 23, 43] that depend on complex secure aggregation techniques due to their requirement for additional non-colluding participants, which alters the original structure of the federated learning framework. Note that the core contribution of this paper is not to propose new model poisoning defense approaches, but to enhance existing popular defenses with privacy features—specifically, server-side similarity-based defenses. Therefore, the experiments aim to evaluate how these privacy-preserving features affect the original defense methods, rather than defending against recent attack techniques and strategies as shown in works like [26, 31, 42, 10, 25].

To assess defense performance, we evaluated the proposed work against popular model poisoning attacks: Inner Product Manipulation (***IPM***) attack [34], ***scaling*** attack[1], and the "a little is enough" (***ALIE***) attack[3]. Unless otherwise mentioned, we assume a default attacker ratio of 0.3 among all participants as malicious clients. The attacks commence at the 50th round and persist until training ends. The default FL training involves 10 clients randomly chosen from 100 for each communication round. Furthermore, we employ a batch size of 64 with each client conducting local training over three epochs per round using an SGD optimizer with a momentum of 0.9 and a learning rate of 0.01. Our *DDFed* implementation's default epsilon ($\varepsilon$) value is set to 0.01 unless specified differently.

### 4.2 Performance Evaluation

**Performance of Defense Effectiveness under Various Attacks.** Figure 2 demonstrates the effectiveness of our *DDFed* method compared to baseline methods in countering three prevalent model poisoning attacks, with an attacker ratio set at 0.3. The attack commences at the 50th round and continues until training concludes. Under the IPM attack scenario, aside from FedAvg, Trimmed Mean, and Clipping Median mechanisms, our approach along with other defense strategies performs well (nearly as model accuracy as without any model poisoning attack) in defending against the IPM attack. The same conclusion also holds true in the ALIE attack. However, only *DDFed* and Clip Median successfully withstand SCALING attacks with minor and acceptable losses in model

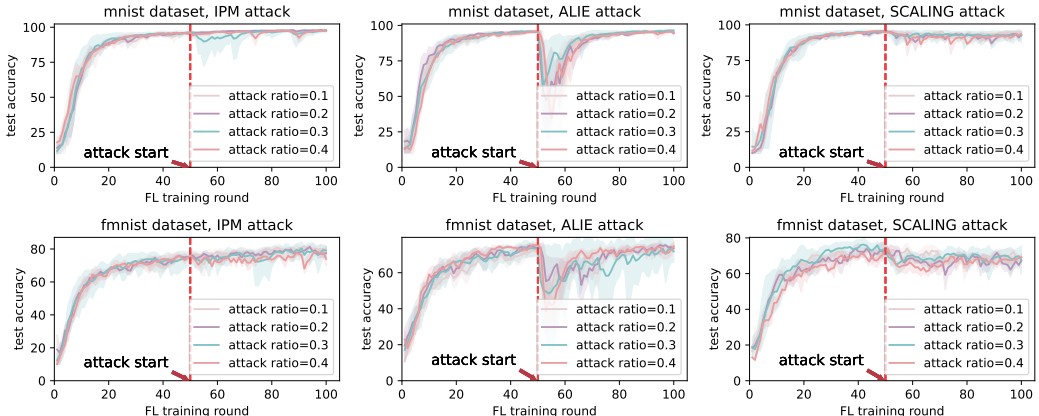

Figure 3: Comparison of *DDFed* effectiveness across different attack ratios, evaluated on MNIST (top) and FMNIST (bottom), under IPM attack (left), ALIE attack (middle), and SCALING attack (right).

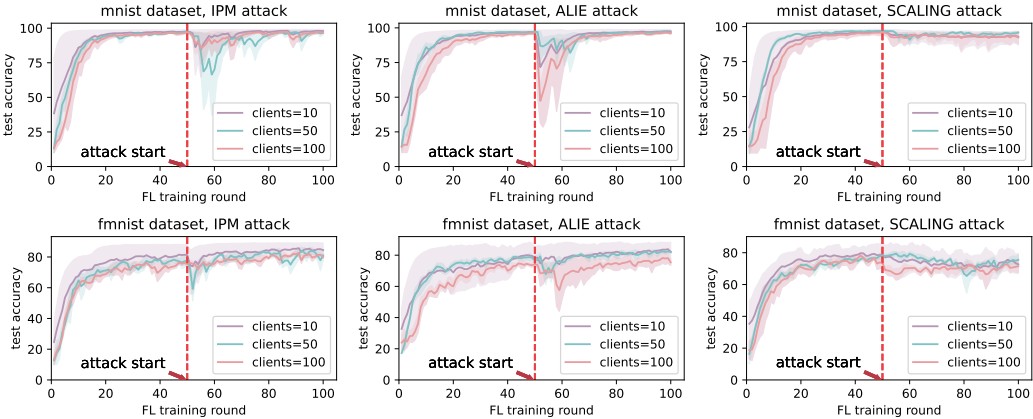

Figure 4: Comparison of *DDFed* effectiveness across different client numbers, evaluated on MNIST (top) and FMNIST (bottom), under IPM attack (left), ALIE attack (middle), and SCALING attack (right).

performance. Note that *DDFed* remains robust even when attackers target the system from the start of training. Due to space constraints, we present the defense effectiveness against various cold-start attacks in Appendix A.2.3. In summary, our *DDFed* method achieves the best comprehensive defense performance.

**Impact of Attacker Ratio.** To further investigate the impact of attacker ratio in the *DDFed* framework, we conducted experiments with various attacker ratio settings. It's important to note that *DDFed* operates under the security assumption that at least half of the participants must be benign (i.e., $r_{attacker} < 0.5$), therefore, in our experiments, the attacker ratio setting is ranged from 0.1 to 0.4. As shown in Figure 3, the proportion of attackers among all clients does not significantly affect our proposed *DDFed* method. This suggests that it can effectively counter three types of model poisoning attacks. Additionally, we observed that under an ALIE attack scenario, our method may require approximately 10-20 training rounds to recover from the continuous attack, depending on the dataset evaluated.

**Compatibility with Cross-device and Cross-silo FL Scenarios.** To explore how the number of clients affects our *DDFed* framework and to confirm its compatibility with two common federated learning scenarios, i.e., cross-device and cross-silo, we conducted multiple experiments. These

experiments had an attacker ratio fixed at 0.3, with client counts varying from 10 to 100. In cross-silo FL, client numbers are typically small, often ranging from a few to several dozen; however, for simulating the cross-device FL scenario in our study, we used 100 clients due to their generally larger population. As illustrated in Figure 4, our *DDFed* framework effectively defends against all three attacks across various client number settings. This suggests that the performance of *DDFed* is not significantly affected by the number of clients, indicating its suitability for both cross-silo and cross-device FL scenarios. Furthermore, a higher number of client settings may result in relatively large fluctuations during training rounds immediately following the attack; however, the model training ultimately converges steadily, unaffected by the continuous attack.

Table 1: Time cost per training round of various defense approaches.

| Approaches | MNIST, IPM attack | | FMNIST, IPM attack | |
|---|---|---|---|---|
| | avg (s) | var (s) | avg (s) | var (s) |
| FedAvg | 10.26 | 0.07 | 10.47 | 0.01 |
| Krum | 10.32 | 0.03 | 10.26 | 0.01 |
| Median | 10.32 | 0.01 | 10.28 | 0.02 |
| Clipping Median | 10.31 | 0.01 | 10.32 | 0.01 |
| Trimmed Mean | 10.32 | 0.02 | 10.30 | 0.01 |
| Cos Defense | 10.25 | 0.01 | 10.26 | 0.02 |
| DDFed (Our Work) | 12.43 | 0.01 | 12.14 | 0.01 |

**Time Cost of Secure Aggregation.**    To assess the additional time cost incurred by integrating FHE-based secure similarity computation and secure aggregation into *DDFed*, we measured the time cost of each training round and compared it with the baseline methods mentioned earlier. All experiments were carried out using the default settings described above. Due to space constraints, we only present the defense approach's time cost per training round when under an IPM attack and have included further results in Appendix A.2.2.

As shown in Table 1, compared to other robust aggregation mechanisms that lack privacy-preserving features, our *DDFed* solution incurs additional time costs due to the integration of FHE-based secure similarity computation and secure aggregation. Across experiments on various datasets and under different attacks, our *DDFed* generally requires an extra 2 seconds compared to the usual 10-second training round, resulting in a 20% increase in time per training round. However, our *DDFed* is capable of defending against model poisoning attacks while also offering strong privacy guarantees. Note that the time-related experiments were conducted on a MacOS platform with an Apple M2 Max chip and 96GB of memory.

**Impact of Epsilon Setting.**    To better understand the effect of the hyperparameter $\varepsilon$ setting on *DDFed*'s perturbation-based secure similarity computation phase, we conducted several experiments with different $\varepsilon$ settings, ranging from 0.01 to 0.1. Here, we only demonstrate the results from 10 clients here, with additional results in Appendix A.2.1.

As shown in Figure 5, the $\varepsilon$ setting has a negligible impact on performance with the MNIST dataset. However, higher $\varepsilon$ values, which indicate stronger DP protection, cause relatively larger fluctuations in performance on the FMNIST dataset. Therefore, we believe that the optimal $\varepsilon$ setting depends on the specific task at hand and leave it as an open question for future research.

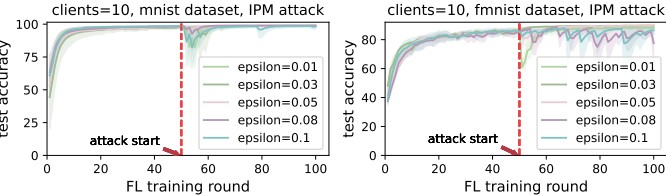

Figure 5: Impact of hyper-parameter $\epsilon$ of differential privacy based perturbation at secure similarity computation phase, evaluated on MNIST (left) and FMNIST (right), under IPM attack.

### 4.3 Discussion and Limitation

To the best of our knowledge, *DDFed* offers a dual defense strategy that simultaneously boosts privacy protection and fights against poisoning attacks in FL, without altering the existing FL framework's architecture. *DDFed* utilizes FHE for top-notch privacy, enabling the aggregation server to perform similarity calculations and aggregation without directly accessing model updates. Additionally, *DDFed* introduces perturbation techniques to block attempts by malicious clients to infer information from similarity scores. It further employs similarity-based anomaly detection, enhanced with strategies like perturbation and post-aggregation clipping, to protect against various types of poisoning attacks. However, *DDFed* has not fully explored two related questions: how can we relax the attacker ratio restriction (i.e., $r_{\text{ATTACK}} < 0.5$) while still ensuring effective dual defense? And how can we adapt *DDFed* to more complex FL scenarios, such as dropout and dynamic participant groups? We leave these questions open for future research. Currently, *DDFed* only enhances existing popular defenses, such as similarity-based strategies with privacy features. Extending *DDFed* to support other or more recent defense strategies remains an open question.

## 5 Conclusion

To tackle the dual challenges of privacy risks and model poisoning in federated learning, we introduce *DDFed*, a comprehensive approach that strengthens privacy protections and counters model poisoning attacks. *DDFed* enhances privacy by using an FHE-based secure aggregation mechanism and addresses encrypted poisoned model detection through an innovative secure similarity-based anomaly filtering method. This method includes secure similarity computation with perturbation and feedback-driven selection process to distinguish safe model updates from potentially harmful ones. Our approach has been rigorously tested against well-known attacks on diverse datasets, demonstrating its effectiveness. We believe our work sets a solid foundation for future advancements in secure and robust federated learning.

## Acknowledgments and Disclosure of Funding

This work is funded by the National Natural Science Foundation of China, under grants No.62302022, No.62225202, No.62202038. We sincerely thank the anonymous reviewers for their insightful comments and constructive feedback, which have greatly improved this paper. Their suggestions were invaluable in refining our analysis and presentation, as well as guiding future research questions related to this work.

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

# A  Appendix

## A.1  DDFed Algorithm

---

**Algorithm 1:** *DDFed* Training

---

**Input:** clients $\{\mathcal{C}_1, ..., \mathcal{C}_m\}$, each client $\mathcal{C}_i$ has its own dataset $D_i$; global training round $T$

**Output:** final global model $\boldsymbol{W}_G^{(T)}$.

1   Each client initializes with FHE key pair $(\text{PK}, \text{SK})$;

2   aggregation server $\mathcal{A}$ initializes the global model $\boldsymbol{W}_G^{(0)}$;

3   **foreach** *training round $t \in \{1, ..., T\}$* **do**

4     **foreach** *client $\mathcal{C}_i \in \{\mathcal{C}_1, ..., \mathcal{C}_m\}$* **do**

5       **if** *not initial training round* **then**

6         $\mathcal{C}_i$ receives $[\![\boldsymbol{W}_G^{(t-1)}]\!]$ from $\mathcal{A}$ and acquires $\boldsymbol{W}_G^{(t-1)} \leftarrow \text{FHE.DEC}_{\text{SK}}([\![\boldsymbol{W}_G^{(t-1)}]\!])$;

7       **if** *reaching final training round $T$* **then**

8         **return** final model $\mathcal{M}_G^{(m)}$;

9       **if** *$\mathcal{C}_i$ is benign* **then** $\mathcal{C}_i$ performs clipping on $\boldsymbol{W}_G^{(t-1)}$;

10       $\mathcal{C}_i$ conducts local training $\boldsymbol{W}_i^{(t)} \leftarrow \text{TRAIN}(\boldsymbol{W}_G^{(t-1)})$;

11       $\mathcal{C}_i$ encrypts local model $[\![\boldsymbol{W}_i^{(t)}, \boldsymbol{W}_i^{L,(t)}]\!] \leftarrow \text{FHE.ENC}_{\text{PK}}([\![\boldsymbol{W}_G^{(t-1)}]\!])$;

12       $\mathcal{C}_i$ sends out $[\![\boldsymbol{W}_i^{(t)}, \boldsymbol{W}_i^{L,(t)}]\!]$ to $\mathcal{A}$;

13     $\mathcal{A}$ waits and collects $\{[\![\boldsymbol{W}_i^{(t)}, \boldsymbol{W}_i^{L,(t)}]\!]\}_{i \in [1,...,m]}$;

14     $\mathcal{A}$ retrieves $[\![\boldsymbol{W}_G^{L,(t-1)}]\!]$ from previous round and prepares perturbation $\Delta^{(t)}$;

15     $\mathcal{A}$ performs $[\![\boldsymbol{s}'^{(t)}]\!] \leftarrow \{\langle [\![\boldsymbol{W}_G^{L,(t)}]\!] + \Delta^{(t)}, [\![\boldsymbol{W}_G^{L,(t-1)}]\!]\rangle\}_{i \in [1,...,m]}$ and sends $[\![\boldsymbol{s}'^{(t)}]\!]$ to $\{\mathcal{C}_1, ..., \mathcal{C}_m\}$;

16     **foreach** *client $\mathcal{C}_i \in \{\mathcal{C}_1, ..., \mathcal{C}_m\}$* **do**

17       $\mathcal{C}_i$ decrypts $\boldsymbol{s}_i'^{(t)} \leftarrow \text{FHE.DEC}_{\text{SK}}([\![\boldsymbol{s}'^{(t)}]\!])$, conducts threshold-based selection and sends back $\boldsymbol{s}_i^{(t)}$;

18     $\mathcal{A}$ collects $\{\boldsymbol{s}_i^{(t)}\}_{i \int [1,...,m]}$ and selects $\boldsymbol{s}^{(t)}$ via majority of votes strategy;

19     $\mathcal{A}$ generates fusion weights $\boldsymbol{f}_{\boldsymbol{s}^{(t)}}^w$ using $\boldsymbol{s}^{(t)}$;

20     $\mathcal{A}$ conducts secure aggregation $[\![\boldsymbol{W}_G^{(t)}]\!] \leftarrow \langle [\![\boldsymbol{W}^{(t)}]\!], \boldsymbol{f}_{\boldsymbol{s}^{(t)}}^{\boldsymbol{W}^{(t)}} \rangle$ and sends $[\![\boldsymbol{W}_G^{(t)}]\!]$ to $\{\mathcal{C}_1, ..., \mathcal{C}_m\}$;

---

The *DDFed* algorithm is outlined in Algorithm 1. Assuming, without loss of generality, that at training round $t$, each client receives the aggregated and encrypted global model from the previous round. Upon decrypting this global model, benign clients clip it before conducting local training. They then encrypt their local model updates after applying a normalization preprocessing step to aid in detecting similarity-based poisoning attacks.

Once all encrypted model updates are collected from the clients, the aggregation server begins secure similarity computations using the abstracted last layer of these updates. It introduces differential privacy by adding perturbation noise and sends them back to each client for collaborative decryption and selection of benign clients.

Following this, based on majority votes, the aggregation server determines final aggregation groups and calculates fusion weights. Finally, it securely aggregates these with the fusion weights to produce an encrypted global model and concludes that round of federated learning (FL) training.

## A.2  Additional Experimental Results

### A.2.1  Impact of Epsilon with 100 Clients

Figure 6 presents the experimental findings on how different $\varepsilon$ values affect perturbations during the secure similarity computation phase, with experiments focusing on an IPM attack scenario and involving 100 clients. These tests were carried out using the MNIST and Fashion-MNIST (FMNIST) datasets. For both datasets, we explored a range of epsilon values from 0.01 to 0.1, noting that lower epsilon values indicate enhanced privacy through increased noise addition. Initially, all configurations demonstrated high accuracy levels; however, performance fluctuations became evident following the attack. Specifically, the MNIST dataset exhibited a notable decrease in accuracy at certain epsilon settings, while the FMNIST dataset showed more moderate variations in performance. Ultimately, both datasets achieved relatively stable model accuracy. Determining the optimal $\varepsilon$ setting is task-dependent and remains an area for future investigation.

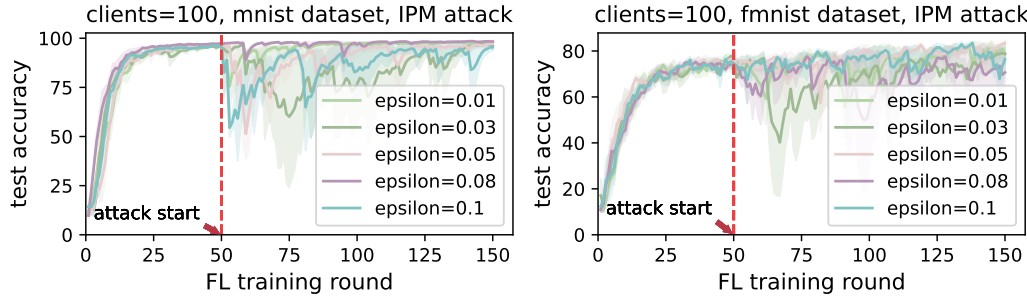

Figure 6: Impact of hyper-parameter $\epsilon$ of differential privacy based perturbation at secure similarity computation phase with client number 100, evaluated on MNIST (left) and FMNIST (right), under IPM attack.

Table 2: Time cost per training round of various defense approaches on MNIST and FMNIST datasets under SCALING attack

| Approaches | MNIST, SCALING attack | | FMNIST, SCALING attack | |
|---|---|---|---|---|
| | avg (s) | var (s) | avg (s) | var (s) |
| FedAvg | 9.87 | 0.21 | 10.51 | 0.01 |
| Krum | 9.75 | 0.21 | 10.42 | 0.01 |
| Median | 9.81 | 0.15 | 10.29 | 0.02 |
| Clipping Median | 9.73 | 0.11 | 10.19 | 0.01 |
| Trimmed Mean | 9.76 | 0.22 | 10.24 | 0.01 |
| Cos Defense | 9.43 | 0.08 | 10.30 | 0.01 |
| DDFed (Our Work) | 12.15 | 0.03 | 12.25 | 0.18 |

### A.2.2 Time Cost of Secure Aggregation on Scaling and ALIE attacks.

Table 2 and Table 3 report additional results on the time cost of each training round taken for various defense strategies against SCALING and ALIE attacks on the MNIST and FMNIST datasets, respectively. Consistent with the findings presented in Section 4.2, our *DDFed* approach adds only 2 seconds to the usual 10-second training round across multiple experiments, datasets, and attack scenarios, resulting in a 20% increase in time per round. Despite this slight increase, *DDFed* successfully defends against model poisoning attacks while ensuring robust privacy protection.

### A.2.3 Performance of DDFed Against Cold-Start Model Poisoning Attacks.

The primary purpose that we initiated the attack at round 50 is to demonstrate the effectiveness of defense mechanisms and clearly show the comparative effects of different defense methods before and after an attack. This setup can also illustrate how various defensive measures impact training convergence and model quality, even without attacks.

*DDFed* is resilient to poisoning attacks from the beginning of training. Our design is not constrained by the attack's initiation round. Supplementary experimental results as reported in Table 4 on the FMNIST dataset with 100 clients in a non-iid setting support this claim.

### A.2.4 Impact of Selected Layer Count on Poisoning Model Detection in DDFed.

In the main body of the paper, we use only the last layer for similarity computation because our primary goal is to integrate privacy-preserving functionality into existing poisoning defense strategies rather than optimizing these mechanisms. Our exploration shows that similarity-based methods and their variants provide comprehensive defense effectiveness, robust against various threat scenarios such as server reliance on validation data, types of model poisoning attacks, and the number of compromised clients. Therefore, we selected a typical similarity-based defense strategy (Cosine Defense) as a starting point to enhance privacy-preserving features. Our approach can easily extend to other similarity-based detection variants using full layers for secure similarity computation. As

Table 3: Time cost per training round of various defense approaches on MNIST and FMNIST datasets under ALE attack

| Approaches | MNIST, ALE attack | | FMNIST, ALE attack | |
|---|---|---|---|---|
| | avg (s) | var (s) | avg (s) | var (s) |
| FedAvg | 10.31 | 0.11 | 10.38 | 0.16 |
| Krum | 10.19 | 0.10 | 10.06 | 0.08 |
| Median | 10.19 | 0.05 | 10.15 | 0.04 |
| Clipping Median | 9.98 | 0.09 | 10.05 | 0.12 |
| Trimmed Mean | 9.97 | 0.11 | 10.06 | 0.06 |
| Cos Defense | 9.78 | 0.17 | 10.14 | 0.09 |
| DDFed (Our Work) | 12.23 | 0.07 | 11.95 | 0.08 |

Table 4: Performance of DDFed Against Cold-Start attacks on FMNIST datasets.

| Approaches | IPM Attacks | ALIE Attacks | SCALINE Attacks |
|---|---|---|---|
| FedAvg | 0 | 10.1 | 0 |
| Krum | 69.05 | 73.69 | 69.95 |
| Median | 67.57 | 76.57 | 74.03 |
| Clipping Median | 61.1 | 73.8 | 75.49 |
| Trimmed Mean | 0 | 43.29 | 0 |
| Cos Defense | 81.87 | 82.97 | 81.11 |
| DDFed (Our Work) | 83.32 | 80.97 | 83.05 |

shown in Table 5, we conducted additional experiments with full-layer secure similarity computation on a larger dataset (CIFAR10) under various attacks.

Table 5: Comparison of Model Performance and Time Cost Across Different Layer Protection Settings on Evaluating the CIFAR10 Dataset with Setting of 60 Training Rounds.

| Approaches | No attack | | IPM attack | | ALIE attack | | SCALINE attack | |
|---|---|---|---|---|---|---|---|---|
| | Acc | Time(m) | Acc | Time(m) | Acc | Time(m) | Acc | Time(m) |
| FedAvg | 70.16 | 46.23 | 0 | 46.62 | 10 | 46.89 | 0 | 46.48 |
| DDFed (Last Layer) | - | - | 70.3 | 50.66 | 64.62 | 51.3 | 69.61 | 51.63 |
| DDFed (Full Layers) | - | - | 69.84 | 58.95 | 69.73 | 58.78 | 68.89 | 59.01 |

## A.3 Differential Privacy

### A.3.1 Differential Privacy

Differential privacy is a mathematical framework designed to provide privacy guarantees for individuals in a dataset. The standard definition of differential privacy is as follows:

A randomized algorithm $\mathcal{M}$ is said to be $(\varepsilon, \delta)$-differentially private if, for any two adjacent datasets $D$ and $D'$ (i.e., datasets differing by only one element), and for any subset of outputs $S \subseteq \text{Range}(\mathcal{M})$, the following inequality holds:

$$\Pr[\mathcal{M}(D) \in S] \leq e^\varepsilon \Pr[\mathcal{M}(D') \in S] + \delta \tag{5}$$

where $\varepsilon$ is the privacy budget parameter, which controls the trade-off between privacy and utility. A smaller $\varepsilon$ indicates stronger privacy. $\delta$ (delta) is a small probability that accounts for the possibility of the privacy guarantee being violated.

The Gaussian mechanism is a specific method to achieve differential privacy by adding Gaussian noise to the output of a function. The definition of the Gaussian mechanism is as follows:

Given a function $f$ and any two adjacent datasets $D$ and $D'$, the sensitivity of $f$ is defined as:

$$\Delta f = \max_{D, D'} \|f(D) - f(D')\|_2 \tag{6}$$

The Gaussian mechanism adds noise drawn from a Gaussian distribution with mean 0 and standard deviation $\sigma$, where $\sigma$ is determined by:

$$\sigma = \frac{\Delta f \sqrt{2 \ln(1.25/\delta)}}{\varepsilon} \tag{7}$$

Thus, the Gaussian mechanism is defined as:

$$\mathcal{M}(D) = f(D) + \mathcal{N}(0, \sigma^2) \tag{8}$$

where $\mathcal{N}(0, \sigma^2)$ denotes a Gaussian distribution with mean 0 and variance $\sigma^2$. By adding Gaussian noise in this manner, the Gaussian mechanism ensures that the output satisfies $(\varepsilon, \delta)$-differential privacy.

### A.3.2 Privacy Analysis of Differentially Private Similarity Computation in *DDFed*

The *DDFed* framework aims to enhance privacy protection and mitigate poisoning attacks within federated learning systems by integrating FHE and a similarity-based anomaly detection system. To further bolster privacy, *DDFed* incorporates DP during the similarity score computation process. This section provides a theoretical analysis of the differential privacy levels maintained by each participant in the *DDFed* framework, specifically focusing on clients during similarity score computation and feedback stages, as well as the aggregation server during model aggregation and similarity score processing.

In the similarity score computation phase, each client normalizes its local model updates before submitting them. To ensure DP, Gaussian noise is added to these normalized updates. By adding Gaussian noise, each client's similarity score computation adheres to $(\varepsilon, \delta)$-differential privacy, ensuring that the privacy of the client's data is preserved even in the presence of adversaries.

During the feedback phase, clients decrypt the similarity scores and submit their results. Since these scores have already been DP due to the added Gaussian noise, the privacy level remains at $(\varepsilon, \delta)$-differential privacy. This ensures that even when clients provide feedback, their privacy is not compromised.

In the model aggregation phase, the aggregation server receives encrypted model updates from clients. While FHE inherently provides a high level of security for these crucial parameters, the aggregation server further ensures privacy by applying DP during the similarity score calculation. The server aggregates the encrypted updates without accessing the plaintext data, thereby maintaining the privacy of the individual model updates.

For the similarity score processing phase, the aggregation server handles the scores submitted by clients, which have already been protected using differential privacy. Consequently, the server does not need to apply additional privacy mechanisms during this phase. The DP guarantees provided during the similarity score computation phase by clients are sufficient to protect the overall process.

Based on the analysis, the privacy levels for each client in *DDFed* framework can be summarized as follows. During the similarity score computation phase, clients achieve $(\varepsilon, \delta)$-differential privacy by adding Gaussian noise to their normalized model updates. During the feedback phase, clients maintain $(\varepsilon, \delta)$-differential privacy as the similarity scores they submit have already been differential private.

By thoughtfully designing and selecting parameters, the *DDFed* framework can provide robust privacy protection and maintain high model performance. The use of FHE for critical parameters and differential privacy for similarity scores ensures a balanced and comprehensive approach to privacy protection, addressing both security and utility needs effectively.

### A.3.3 Impact of DP on FHE-based Similarity Computation in *DDFed*

Generally, the reader may concern about whether $[[x]] + \Delta$ equals $x + \Delta$, where $x$ is under FHE protection. However, this depends on the precision of the employed FHE schemes. Proving such a statement theoretically may require delving into the specific construction algorithm of the FHE scheme, which is beyond the scope of machine learning-oriented venues.

Table 6: Impact of DP on FHE-based Similarity Detection in DDFed on evaluating CIFAR10 datasets.

| Approaches | IPM Attacks | ALIE Attacks | SCALINE Attacks |
|---|---|---|---|
| DDFed (Simulated) | 70.21 | 64.3 | 69.82 |
| DDFed (Our Work) | 70.31 | 64.62 | 69.6 |

This paper utilizes CKKS constructions, which natively support high-precision secure computation on floating-point numbers. As a result, adding DP noise to encrypted similarity results does not degrade performance. To validate this, we conducted supplementary experiments on CIFAR10 using a simulated DDFed setup where DP noise was added to non-encrypted parameters. The reported results in Table 6 support this claim.

