# OpenReview forum: "Dual Defense: Enhancing Privacy and Mitigating Poisoning Attacks in Federated Learning"
_NeurIPS.cc/2024/Conference — NeurIPS 2024 poster_

### Official Review · Reviewer_VKjn · 2024-06-30

**Soundness:** 3
**Presentation:** 3
**Contribution:** 3
**Rating:** 5
**Confidence:** 3

**Summary:**

This paper introduces DDFed, which combines FHE (F8lly Homomorphic Encryption) and similarity computation/collaborative selection, to achieve privacy protection and attack mitigation, respectively.  The idea of this paper is to leverage FHE within Federated Learning (FL) as a strategy for protecting privacy while addressing the problems it introduces in mitigating attacks.

**Strengths:**

Originality: the paper's approach to solving the dual challenges with a singular framework is an innovative aspect that extends the current conversation within the field.

Quality: the results appears to be robust, with clear experimental setups and codes. The use of standard datasets such as MNIST and FMNIST ensures that the findings are comparable and relevant to existing work

Clarity: this paper is well-organized and written in a manner that is accessible to readers familiar with federated learning and cryptographic methods.

**Weaknesses:**

I am somewhat concerned about the practicality of the method described in the paper, as well as the practicality of some settings, See Questions.

**Questions:**

I appreciate the author's dual focus and have also reviewed the code provided.
However, I require further clarification before considering the paper for acceptance.
First, in line 186, to circumvent division operations, DDFed moves the comparing tasks to the client side, but why should this task be calculated on the client side if some of the clients are malicious?

Second, the paper demonstrates that DDFed incurs higher training costs compared to other methods. It reports an average training round duration exceeding 12 seconds on small datasets like MNIST and FMNIST, which are only 28x28 in size. Moreover, while FHE typically involves significant communication overhead, the paper lacks detailed experimental results or a discussion on this aspect. Given that communication costs can rise exponentially with increasing model parameters, the practicality of this method is questionable.

**Limitations:**

See questions.

---

> ### Author Response · Authors · 2024-08-06
>
> We appreciate the reviewer's comments and the raised concerns.
>
> **Resp to Q1:**
> DDFed moves the comparison task to the client side because no FHE scheme efficiently supports both comparison operations and floating-point numerical computation over encrypted model updates. FHE schemes are generally classified into three categories: bit-wise approaches like FHEW and TFHE, word-wise approaches like BGV and BFV, and CKKS for efficient floating-point computations. There is currently no perfect FHE solution that efficiently handles all types of computational operations. Each category is more efficient for specific applications, leading to trade-offs when employing a particular FHE scheme.
>
> In this paper, we use FHE to protect FL local models with millions of floating-point parameters, prioritizing computational efficiency. We adopt the CKKS scheme as our underlying FHE solution and delegate comparison tasks to the client side since CKKS does not efficiently support comparison operations.
>
> We encountered issues with potential malicious clients in our initial DDFed design, as noted by the reviewer. This prompted us to propose a feedback-driven collaborative selection mechanism. Similar to blockchain consensus technology, this mechanism can tolerate up to 50% of clients being malicious. This aligns with our threat model assumption that DDFed can handle adversaries with the ratio lower than 0.5.
>
>
> **Resp to Q2:**
> The introduction of FHE for privacy-preserving federated learning often raises computational concerns due to secure computation over encrypted models and communication overhead from the larger size of encrypted model updates, as noted by the reviewer.
>
> From a computational perspective, the cost of FHE for privacy enhancement in FL depends on the size of the trained model, not the dataset. In our experiment, evaluating models on the MNIST and FMNIST datasets involved about 0.23 million parameters. Additionally, as shown in our released prototype implementation, we run each training round sequentially with each client rather than in parallel. We believe that if DDFed is applied to a real FL scenario where all local training and operations are conducted in parallel, the total training time will be lower than reported in Table 1.
>
> The primary goal of this paper is to validate the effectiveness of the proposed dual defense approaches. As indicated in the released open-source code, we implemented a simulated FL framework rather than an actual FL system at this prototype stage. Building an FL system in a distributed real-world network environment slightly deviates from the core focus of this paper. Therefore, we did not evaluate communication overhead by measuring network latency caused by transferring encrypted model updates instead of plaintext model updates.
>
> Formally, depending on the security parameter settings, the communication overhead is linearly related to the model size, rather than an exponential relationship. In our CKKS setting with a security parameter of 128, we use the above-mentioned model with 0.23 million parameters as an example. Our manual measurement shows that the original model size is approximately 0.9MB, while the encrypted model size is about 20.4MB. This indicates that the communication payload overhead is roughly 20 times greater compared to a non-FHE protected solution.
>
> In summary, our work may introduce additional computational and communication costs similar to most existing FHE-based privacy-preserving federated learning solutions. However, this design provides a strong guarantee of privacy preservation. While applying DDFed in a cross-device FL scenario involving thousands of devices might raise practical concerns, we believe it remains feasible for cross-silo FL scenarios where organizations or companies with higher network bandwidth and servers conduct FL training. It is worth noting that the cross-silo FL scenario is an important type of application according to recent surveys on federated learning.
>
> Again, we appreciate the reviewer's comments. We hope our response addresses these concerns and positively influences the final decision with higher rating scores.

---

### Official Review · Reviewer_zJWB · 2024-07-12

**Soundness:** 3
**Presentation:** 3
**Contribution:** 3
**Rating:** 6
**Confidence:** 4

**Summary:**

This paper proposes a novel Byzantine-robust and differentially private federated learning (FL) framework, named as Dual Defense (DDFed). To guarantee the privacy, DDFed utilizes a secure similarity computation based on fully homomorphic encryption, without leaking client’s privacy to either the server or any potential compromised clients. To address the Byzantine attacks, DDFed uses a novel feedback-driven collaborative selection method to filter out malicious clients by majority of votes.

**Strengths:**

1. This paper proposes a novel and efficient Federated Learning (FL) framework that addresses both privacy and Byzantine robustness issues without compromising the performance of the trained model.
2. DDFed uses a carefully designed feedback-driven collaborative selection method that allows clients to participate in detection while excluding the influence of differential privacy (DP) noise from the aggregated weights.
3. Experiments on different datasets with non-IID data setting illustrate the effectiveness of DDFed.

**Weaknesses:**

1. DDFed only considers the last layer in the detection process. Although this approach makes DDFed more efficient compared to other methods, it is vulnerable to attackers who can bypass DDFed by injecting poisoned weights into other hidden layers.
2. The authors add differential privacy (DP) noise to the encrypted parameters in a manner commonly used for non-encrypted parameters. They should provide either theoretical proof or experimental evidence to verify that the DP noise remains effective when the parameters are decrypted.
3. The authors should extend their experiments. For instance, they should use more complex datasets (e.g., CIFAR-10) and larger models (e.g., ResNet). Additionally, they should compare DDFed with more advanced attacks [1] and defenses (e.g., Multi-Krum and Flame [2]). Furthermore, the authors should demonstrate the robustness of DDFed even when the attacker targets the model from the beginning of the FL training.
4. The authors should better discuss DDFed with previous methods that do not use fully homomorphic encryption (FHE) and sophisticated differential privacy (DP) noise utilization [3-5], further illustrating the strengths of DDFed.

[1] Fang, Minghong, et al. "Local model poisoning attacks to {Byzantine-Robust} federated learning." 29th USENIX security symposium (USENIX Security 20). 2020.

[2] Nguyen, Thien Duc, et al. "{FLAME}: Taming backdoors in federated learning." 31st USENIX Security Symposium (USENIX Security 22). 2022.

[3] Miao, Yinbin, et al. "Privacy-preserving Byzantine-robust federated learning via blockchain systems." IEEE Transactions on Information Forensics and Security 17 (2022): 2848-2861.

[4] Guo, Hanxi, et al. "SIREN+: Robust Federated Learning with Proactive Alarming and Differential Privacy." IEEE Transactions on Dependable and Secure Computing (2024).

[5] Liu, Xiaoyuan, et al. "Privacy-enhanced federated learning against poisoning adversaries." IEEE Transactions on Information Forensics and Security 16 (2021): 4574-4588.

**Questions:**

1. Why the authors choose to start the attack after round 50? Will DDFed still be robust when the attackers attack the global model from the beginning of the training?
2. Why is only the last layer considered in the secure similarity computation? Attackers could easily bypass such detection by injecting poisoned weights into other hidden layers.
3. Will FHE influences the privacy protection of DP noise? Please also refer to the second Cons in the Weakness section.
4. Is the cosine similarity between the weights a good indicator? The attacker could add restrictions to the poisoned parameters to influence the global model without flip the cosine similarity.

**Limitations:**

The authors should further clarify the limitations of the proposed method. Specifically, they should address the fact that it only considers the last layer in the detection process and adds DP noise to encrypted data in the same manner as plain data. If the author could well address my concerns from theses two perspectives, I will give a higher score.

---

> ### Author Response · Authors · 2024-08-06
>
> We appreciate the reviewer's concerns and suggestions.
>
> **Resp to Q1:**
> The primary purpose that we initiated the attack at round 50 is to demonstrate the effectiveness of defense mechanisms and clearly show the comparative effects of different defense methods before and after an attack. This setup can also illustrate how various defensive measures impact training convergence and model quality, even without attacks.
>
> DDFed is resilient to poisoning attacks from the beginning of training. Our design is not constrained by the attack's initiation round. Supplementary experiments on the FMNIST dataset with 100 clients in a non-iid setting support this claim.
>
> |Approaches|Attacks Type|Acc|
> |--|--|--|
> |FedAvg|IPM|0|
> ||ALIE|10.1|
> ||SCALINE|0|
> |Krum|IPM|69.05|
> ||ALIE|73.69|
> ||SCALINE|69.95|
> |Median|IPM| 67.57|
> ||ALIE| 76.57|
> ||SCALINE|74.03|
> |Clip Median|IPM|61.1|
> ||ALIE|73.8|
> ||SCALINE|75.49|
> |Trimmed Mean|IPM|0|
> ||ALIE|43.29|
> ||SCALINE|0|
> |Cosine Defense|IPM|81.87|
> || ALIE| 82.97|
> || SCALINE| 81.11|
> |DDFed (Our work)|IPM|83.32|
> ||ALIE|80.97|
> ||SCALINE|83.05|
>
> **Resp to Q2:**
> We use only the last layer for similarity computation because our main goal is to add privacy-preserving functionality to existing poisoning defense strategies, more than on optimizing existing defense mechanisms.
> Based on our exploration, similarity-based methods and their variants offer a comprehensive defense effectiveness that is robust to various threat assumption setting such as server reliance on validation data, types of model poisoning attacks, number of compromised clients. Therefore, we chose one typical similarity-based defense strategy (Cosine Defense) as a starting point to enhance privacy-preserving features. Our approach can easily extend to other similarity-based detection variants that use full layers for secure similarity computation. We conducted additional experiments considering full-layer secure similarity computation on a larger dataset (CIFAR10) under various attacks.
>
> |Approaches|Attacks Type|Acc|Time Cost (min) (60 rounds)|
> |--|--|--|--|
> |FedAvg|NO ATTACK|70.16|46.23|
> ||IPM|0|46.62|
> ||ALIE|10|46.89|
> ||SCALINE|0|46.48|
> |DDFed (Last Layer)|IPM|70.3|50.66|
> ||ALIE|64.62|51.3|
> ||SCALINE|69.61|51.63|
> |DDFed (Full Layers)|IPM|69.84|58.95|
> ||ALIE|69.73|58.78|
> ||SCALINE|68.89|59.01|
>
> **Resp to Q3:**
> In the appendix 3, we provided an overview privacy analysis of differential privacy-enhanced FHE-based secure similarity computation but did not include a formal proof. Generally, we believe the reviewer is concerned about whether $[[x]]+ \Delta$ equals $x + \Delta$, where $x$ is under FHE protection. However, this depends on the precision of the employed FHE schemes. Proving such a statement theoretically may require delving into the specific construction algorithm of the FHE scheme, which is beyond the scope of machine learning-oriented venues.
>
> This paper utilizes CKKS constructions, which natively support high-precision secure computation on floating-point numbers. As a result, adding DP noise to encrypted similarity results does not degrade performance. To validate this, we conducted supplementary experiments on CIFAR10 using a simulated DDFed setup where DP noise was added to non-encrypted parameters. The reported results support this claim.
>
> |Approaches|Attacks Type|Acc|
> |--|--|--|
> |DDFed (Simulated)|IPM|70.21|
> ||ALIE|64.3|
> ||SCALINE|69.82|
> |DDFed(Our work)|IPM|70.31|
> ||ALIE|64.62|
> ||SCALINE|69.6|
>
> **Resp to Q4:**
> As stated in response 2, the primary goal and contribution of this paper are to enhance existing model poisoning defense strategies with privacy preservation functionality, addressing both privacy and model poisoning issues in FL simultaneously. Therefore, we did not focus heavily on indicator selection.
>
> Our exploration in this field shows that similarity-based model poisoning defense methods and their variants provide comprehensive defense effectiveness when considering threat model assumptions such as server reliance on validation data, types of model poisoning attacks, number of compromised clients, and attacking rounds. While it may not be optimal for a specific threat model assumption, it offers relatively good overall defense quality as demonstrated by related papers.
>
> This paper focuses more on privacy-preserving functionality than on model poisoning defense strategies. Therefore, we did not conduct the latest attacks [1-2] for comparison, as suggested by the reviewer, because it is outside the focus of this study.
>
> **Supplementary response:**
> We also conducted additional experiments to evaluate our work on the CIFAR10 using a ResNet model as shown in ***Resp to Q2*** section. Regarding related work, we have discussed the referenced study [5], but omitted studies [3-4]. These will be included in the final version.
>
> Finally, we hope our response, particularly regarding the last layer setting and DP noise addition, addresses these concerns and positively influences the final decision.

---

> > ### Comment · Reviewer_zJWB · 2024-08-08
> > **Thanks for the clarification**
> >
> > Your rebuttal has addressed my first concern regarding the last layer setting and has partially resolved my concern about the DP noise addition, so I will raise my score to 5.
> >
> > However, as mentioned in my review, I still recommend that the authors evaluate the robustness of DDFed against more advanced attacks (e.g., adaptive attacks) and assess the DP performance using a simple inversion attack (if the theoretical proof is time-consuming). If the authors can further address these concerns, I will consider raising my score again.
> >
> > P.S. It seems that the authors did not correctly set the readers of the rebuttal, preventing me from accessing the general response as well as the rebuttal to other reviewers.

---

> ### Author Response · Authors · 2024-08-11
>
> **Resp to advance attacks**:
> We thoroughly examined the advanced attack suggested by the reviewer, as referenced in [1] above. That paper only discusses attacks on Krum and its variant, Trimmed Mean, and Median—all of which are not conventional similarity-based approaches. Therefore, we adopted a similar attack strategy and implemented a dynamic attack: conducting n attack iterations after n non-attack iterations to collect "before-attack" global models. We then averaged the crafted models (using the most effective attack) with the "before-attack" global models for subsequent periods of attack iterations. The results on CIFAR10 show that DDFed remains effective under such dynamic attacks.
>
> | Approaches       | Attacks Type | Acc   |
> | ---------------- | ------------ | ----- |
> | FedAvg           | NO ATTACK    | 70.16 |
> | DDFed (Our work) | IPM          | 70.31 |
> |                  | ALIE         | 64.62 |
> |                  | SCALINE      | 69.6  |
> |                  | Dynamic ALIE | 64.52 |
>
> **Resp to DP protection and evaluation**:
> We would like to emphasize that the DP-based perturbation is used to prevent potential privacy leakage from similarity scores. Specifically, it prevents adversaries from inferring private information about benign clients by exploiting decrypted similarity scores and previous global models, rather than directly targeting a specific client's model update as in the previous DGL attack.
>
> As shown in formula 4 of the paper, suppose a client can infer private information; they must (i) solve a multivariate linear equation problem with only one equation successfully—given $C$, $B$, and $C=<A,B>$, find all parameters of $A$—and (ii) break the DP-based perturbation to find $W^{t-1}_i$. To our knowledge, we do not believe an adversary can accomplish step (i) because each round's $A$ is not fixed. If the similarity computation does not use full layers, this challenge becomes even greater.
>
> However, let's consider a harsh condition: during convergence phases when model parameters tend to stabilize and A is approximately fixed while using full-layer models for similarity computing. We then conducted a simple inversion attack (e.g., DLG attack) as suggested by the reviewer. We assessed the PSNR results of both DP-based perturbation and non-DP perturbation on the MNIST dataset. The results below demonstrate the effectiveness of the DP-based similarity score perturbation.
>
> | Approaches         | Attacks    | PSNR (avg of last 10 round) |
> | ------------------ | ---------- | --------------------------- |
> | DP Perturbation    | DLG Attack | 28.5                        |
> | No DP Perturbation | DLG Attack | 18.2                        |
>
> **By the way**, we didn't set up special access control for readers; we just used the default settings when inputting the last round's response. We also haven't provided a general response yet for that round. As suggested, we've revised the scope of previous responses and added a general response to address the current status of the rebuttal.
>
> Finally, we appreciate the reviewer's concerns and suggestions again. We hope our response has addressed these issues and will continue to positively influence the final decision in this round of discussions.

---

> > ### Comment · Reviewer_zJWB · 2024-08-12
> > **Thanks for the reply.**
> >
> > Thank you for the response, but there's still a point of confusion in your results: How did you compute the PSNR? Typically, a higher PSNR indicates a more successful inversion, so it's surprising that the DP-based version has a much higher PSNR than the non-DP version.
> >
> > The reason I requested the inversion attack results is that it could serve as complementary evidence to support your DP+HE scheme, alongside the theoretical proof. Specifically, I expected a comparison of three schemes:
> > 1. No DP, No HE (baseline)
> > 2. With DP, No HE (previous DP-based method)
> > 3. With DP, With HE, and DP noise on encrypted data (your method)
> >
> > If your method's results show comparable defense effectiveness to the traditional DP-based method, it would strongly support the validity of adding DP noise to encrypted data.

---

> > > ### Author Response · Authors · 2024-08-14
> > >
> > > Thank your for the further comments.
> > >
> > > PSNR is calculated using the Mean Squared Error (MSE) between the original and reconstructed images. We just realized that we made a mistakes in the above reported table due to limited response time and our negligent. DP-based version actually has a lower PSNR than the non-DP version.
> > >
> > > Regarding the mentioned three comparison solution,
> > > - Scheme 1 is actually the Cosine Defense solution. Without HE, it does not provide privacy-preserving functionality, resulting in any inversion attack works well as they attack on native FL. It is reported above, where PSNR is 28.5.
> > > - Scheme 2 is reported above, where the PSNR is actually 18.2.
> > > - Scheme 3 is our work. As we give a short theoretical analysis above, according to most HE related work, it actually does not impact the computing precision in the context of FL. Due to limited response time, we cannot provide detailed experiments right now. We will include those experimental results in the final version.
> > >
> > > Finally, we appreciate the reviewer's further comments again. We hope our response has addressed these issues and will continue to positively influence the final decision in this round of discussions.

---

> > > > ### Comment · Reviewer_zJWB · 2024-08-14
> > > > **Thanks for the clarification. I've raised my score to 6.**
> > > >
> > > > Thank you for your clarification. Please ensure that you include the experimental results and references discussed during the rebuttal in the next version of the main text, as they are crucial to the soundness of your paper. My concerns have been largely addressed, so I further raise my score to 6. Good luck!

---

### Official Review · Reviewer_65Lv · 2024-07-13

**Soundness:** 2
**Presentation:** 2
**Contribution:** 2
**Rating:** 3
**Confidence:** 4

**Summary:**

This paper introduces a Dual Defense Federated learning (DDFed) framework. DDFed simultaneously boosts privacy protection and mitigates poisoning attacks leveraging fully homomorphic encryption (FHE). The experiments with publicly accessible datasets demonstrate DDFed’s effectiveness in safeguarding model privacy and robustly defending against model poisoning threats.

**Strengths:**

- This paper studies an important and instereting problems of dual defense.

- The presentation is clear and easy-to-follow.

**Weaknesses:**

Major concerns:

- The methods fails to address the non-colluding problem. As the clients hold the same private key, the server would easily obtain the private gradient if it collaborate with one of the client.

- The robust aggregation protocol relies on the cosine similarity with the global model from previous round, raising two problem: 1) why do the author select this method as the aggregation rule? Could the framework extend to other aggregation protocols? Please include discussion for the choice and comparison of defense approaches.  2) The protocol relies on the assumption that the inital round consists of all benign clients. How to ensure the condition?

Minor concerns:

- The experiments uses two simple dataset, MNIST and Fashion-MNIST. Please consider more complex dataset and models.

- Table 1 shows that the training time takes only 2 more seconds than FedAvg. The result is quite conter-intuitive as the FHE takes significant time for aggregation and there's no improvement on the FHE algorithm itself. How many clients are involved in the evaluation? Please consider more clients (>100) and larger model size.

**Questions:**

Refer to weakness.

**Limitations:**

The authors discuss the limitation.

---

> ### Author Response · Authors · 2024-08-06
>
> We appreciate the reviewer's concerns and suggestions.
>
> **Resp to major concern 1:**
> Collusion between the server and clients is beyond the scope of our threat model assumptions in Section 3.1. In PPFL, each solution includes a threat model that defines the adversary's capabilities and behavior, following the security research tradition of defining threat boundaries before proposing solutions since no solution can claim absolute security.
> Our threat model assumes an honest-but-curious server that does not collude with any clients, consistent with most related work's assumptions. Most current PPFL approaches cannot handle such collusion issues, which is a strong security assumption.
> We believe our work represents substantial progress in this field. Our solution eliminates the need for a non-colluding two-server assumption found in most recent related works, making private and robust FL solutions more practical.
>
> **Resp to major concern 2:**
> Based on our exploration in model poisoning attacks and defenses, our findings indicate that similarity-based detection methods and their variants provide excellent and comprehensive defense outcomes regarding the server's reliance on validation data, types of model poisoning attacks, number of compromised clients, and attacking round. This is why we have chosen the (cosine) similarity-based poisoning model detection methodology.
> Additionally, our approach can easily extend to similarity-based poisoning model detection variants. However, it does not provide a unified support to all other defense methods such as Krum or those relying on the assumption that the server has partial validation datasets. The reviewer's question represents a promising future direction. Regarding comparisons with other defense approaches, we have demonstrated their effectiveness in existing experimental evaluations, such as in Figure 2. We will include discussions of these choices in the final version.
>
> Besides, our solution does not rely on the assumption that the initial round of training must include only benign clients. In fact, our work supports starting attacks from the first training round, meaning compromised clients can be present initially. To substantiate this claim, we conducted additional experiment, comparing our approach with baselines by initiating attacks in the first training round on the FMNIST with 100 clients in a non-iid setting (see reported results below).
> The reviewer's misunderstanding likely arises from our experimental setup and results. We began the attack at round 50 to demonstrate the effectiveness of defense mechanisms and to clearly show the comparative effects of different defense methods before and after an attack. This helps illustrate how various defensive measures impact training convergence and model quality, even in the absence of attacks.
>
> |Approaches|Attacks Type|Acc|
> |--|--|--|
> |FedAvg|IPM|0|
> ||ALIE|10.1|
> ||SCALINE|0|
> |Krum|IPM|69.05|
> ||ALIE|73.69|
> ||SCALINE|69.95|
> |Median|IPM| 67.57|
> ||ALIE| 76.57|
> ||SCALINE|74.03|
> |Clip Median|IPM|61.1|
> ||ALIE|73.8|
> ||SCALINE|75.49|
> |Trimmed Mean|IPM|0|
> ||ALIE|43.29|
> ||SCALINE|0|
> |Cosine Defense|IPM|81.87|
> || ALIE| 82.97|
> || SCALINE| 81.11|
> |DDFed (Our work)|IPM|83.32|
> ||ALIE|80.97|
> ||SCALINE|83.05|
>
> **Resp to minor concern 1:**
> Given the limited response time, we conducted an additional experiment to evaluate our approach against related baselines on the CIFAR10 using a ResNet model with 100 clients in a non-iid setting. As shown in the tables below, our proposed method remains effective under various attacks.
>
> |Approaches|Attacks Type|Acc|
> |--|--|--|
> |FedAvg|NO ATTACK|70.16|
> ||IPM|0|
> ||ALIE|10|
> || SCALINE|0|
> |Cosine Defense|IPM|68.04|
> ||ALIE|68.27|
> || SCALINE|69.05|
> |DDFed (Our work)|IPM|70.31|
> ||ALIE|64.62|
> ||SCALINE|69.6|
>
> **Resp to minor concern 2:**
> Using HE for privacy enhancement does not add significant overhead in FL recently, contrary to what most non-cryptography researchers might expect. Despite the inherent challenges of developing HE technology, the minor computational overhead reported in Table 1 is due to the following factors:
> 1) The scale of HE used is relatively small, with total parameters of models being about 0.23 million.
> 2) Our design mechanism involves only simple computations over the encrypted model, with just one layer of multiplication depth. It is well-known that multiplication depth significantly impacts efficiency of HE.
> As suggested, we conducted additional experiments on the CIFAR10 dataset using a ResNet model (approximately 11.2 million parameters) and 100 clients. The results showed that it takes an extra 974 seconds (16 minutes) compared to the non-protected FedAvg baseline. This increase is reasonable and acceptable given that our proposed dual defense approach offers strong privacy and security guarantees.
>
> We appreciate the reviewer's concerns and suggestions. We hope our response addresses these concerns and influences the final decision positively.

---

> ### Comment · Reviewer_65Lv · 2024-08-10
>
> Thanks for the response and additional experiments. However, some of my concerns remains.
>
> **Collusion between clients and server**: Firstly, assuming no collusion between clients and servers poses great vulnerabilities in real-life, particularly with expanding client pools. Servers can easily introduce fake clients to undermine this assumption. Recent studies [1][2] have successfully addressed privacy concerns and ensured robust aggregation, eliminating the need of this assumption. Secondly, the paper mentioned by the authors [3] relies on two servers to validate the gradient normalization. The authors remove the two-server setting at the cost of simply skipping the normalization step. How to ensure that the clients' local weights are normalized?
>
> **Choise of similarity-based detection methods**: I understand that the solution does not rely on the assumption that the initial round of training must include only benign clients. However, the aggregation protocol the authors choose does not have formal convergence guarantee. As is mentioned by another reviewer, I expect the theoretical proof under the case with and without DP noise.
>
> [1] Lycklama, H., Burkhalter, L., Viand, A., Küchler, N., & Hithnawi, A. (2023, May). Rofl: Robustness of secure federated learning. In 2023 IEEE Symposium on Security and Privacy (SP) (pp. 453-476). IEEE.
>
> [2] So, J., Güler, B., & Avestimehr, A. S. (2020). Byzantine-resilient secure federated learning. IEEE Journal on Selected Areas in Communications, 39(7), 2168-2181.
>
> [3] Ma, Z., Ma, J., Miao, Y., Li, Y., & Deng, R. H. (2022). ShieldFL: Mitigating model poisoning attacks in privacy-preserving federated learning. IEEE Transactions on Information Forensics and Security, 17, 1639-1654.

---

> > ### Author Response · Authors · 2024-08-11
> >
> > **Resp to collusion concern**:
> > First, we argue that the assumption of client-server collusion is uncommon and does not affect practicality in this field for the following reasons:
> > 1. While assuming no client-server collusion might introduce vulnerabilities, it is an extreme case in real-world applications. As noted by the reviewer, if a server can actively attack the FL system (e.g., by introducing fake clients), it could infer more private information, but easily bypass server-side defenses against model poisoning attacks due to its ability to break aggregation rules. Typically, the threat model assumption involves a trade-off and does not aim to cover all extreme cases.
> > 2. Assuming no client-server collusion remains practical for real-world applications. FL is typically classified into cross-silo FL and cross-device FL from an application perspective. In cross-silo scenarios, servers are usually authoritative organizations within or between industries. In cross-device scenarios, cloud service providers often act as servers. In these cases, servers are generally honest-but-curious rather than active attackers. We acknowledge that the no-client-server-collusion assumption may not cover all scenarios but believe it is still reasonable and practical for most applications since there is no one-size-fits-all solution.
> > 3. To our knowledge, most existing papers—may be approximately 99%—do not consider client-server collusion. For instance, one of the referenced papers [1,2] does not assume such collusion either; paper [2], which shares similar assumptions with ours, explicitly states that it does not address client-server collusion.
> > In summary, our work aligns with the majority of existing solutions and we believe these solutions have their respective application value in specific scenarios.
> >
> > Second, our work does not actively ensure that clients' local weights are normalized. In fact, non-normalized local weights will deviate more from the global model, making them easier to detect and filter using similarity-based approaches.
> > In DDFed, a benign client will correctly follow the protocol and generate normalized local weights accurately. If the client is adversarial, it may or may not perform normalization; however, this does not provide an advantage in circumventing the defense method.
> >
> > **Resp to concerns of choosing similarity-based detection method**:
> > We did not include the formal convergence guarantee in the current version for two reasons:
> > 1. The convergence guarantee has already been proven in previous similarity-based solutions. Our work follows this method, enhancing it with privacy-preserving functionality using our proposed solution, without changing the aggregation rule. Therefore, covering this is not a contribution of our paper.
> > 2. Differential Privacy (DP) perturbation is only applied to secure similarity computation and does not affect the secure aggregation stage, hence it does not impact convergence.
> > Additionally, as noted by the reviewer, our work relates to ShieldFL. We use a similar similarity-based detection approach but employ different privacy-enhancing technologies. Thus, the formal convergence is akin to that in ShieldFL. We will include an analysis of the formal convergence guarantee in the appendix of the final version.
> >
> >
> > Regarding the concern about DP-based perturbation, we may not have correctly implemented the reader access control policy in our previous response. Actually, we initially addressed reviewer zJWB's concerns and adjusted the reader policy so you can track all other responses. For your convenience, here is our response to this concern raised by reviewer zJWB.
> >
> > In Appendix Section 3, we provided an overview of privacy analysis for differential privacy-enhanced FHE-based secure similarity computation but did not include a formal proof. Generally, we believe the reviewer's concern is whether $[[x]] + \Delta$ equals $x + \Delta$, where $x$ is under FHE protection. This depends on the precision of the employed FHE schemes. Proving this theoretically would require delving into the specific construction algorithm of the FHE scheme, which is beyond our scope.
> > This paper uses CKKS FHE constructions that natively support high-precision secure computation on floating-point numbers. Therefore, adding DP noise to encrypted similarity results does not degrade performance.
> > As suggested by reviewer zJWB, we conducted supplementary experiments on CIFAR10 using a simulated DDFed setup where DP noise was added to non-encrypted parameters. The reported results support this claim.
> >
> > |Approaches|Attacks Type|Acc|
> > |--|--|--|
> > |DDFed (Simulated)|IPM|70.21|
> > ||ALIE|64.3|
> > ||SCALINE|69.82|
> > |DDFed(Our work)|IPM|70.31|
> > ||ALIE|64.62|
> > ||SCALINE|69.6|
> >
> > We invite the reviewer to continue following our response thread with reviewer zJWB for further updates.
> >
> > Again, we appreciate the reviewer's suggestions. We hope our response addresses these concerns and influences the final decision positively.

---

> > > ### Comment · Reviewer_65Lv · 2024-08-14
> > >
> > > Thanks for your reply. I still have the following comments:
> > >
> > > > To our knowledge, most existing papers—may be approximately 99%—do not consider client-server collusion.
> > >
> > > Actually, SecAgg[1][2] algorithm has rigorous security/threshold analysis for the client-server collusion. Since one of the paper's major contribution is to solve the non-colluding problem, it might be insufficient to consider merely the two-server collusion (the more reliable one) while ignoring the client-server collusion (the more vulnerable one). Enhancing your framework to include the collusion threshold between clients and servers would strengthen its impact.
> > >
> > > > Second, our work does not actively ensure that clients' local weights are normalized. In fact, non-normalized local weights will deviate more from the global model, making them easier to detect and filter using similarity-based approaches.
> > >
> > > This argument sounds inaccurate. If $\|w_i\|$ is greater than 1, then un-normalizing in fact makes $cos(\alpha_i)$ greater. That it, skipping the normalization would put the adversaries at an advantage.
> > >
> > > > The convergence guarantee has already been proven in previous similarity-based solutions.
> > >
> > > The similarity-based methods with convergence guarantee differs from your methods. For example, in ShieldFL, they use the similarity between current gradient and previous gradients to filter outliers (instead of the cosine simlarity between local and global weights). Your aggregation method follows that proposed by [3], but they didn't provide convergence guarantee.
> > >
> > > > Differential Privacy (DP) perturbation is only applied to secure similarity computation and does not affect the secure aggregation stage, hence it does not impact convergence....We will include an analysis of the formal convergence guarantee in the appendix of the final version.
> > >
> > > I understand that DP perturbation provides strong privacy guarantee, and that the DP noises are not added to the aggregation results. However, it would impact the selection of benign clients. I look forward the analysis of the formal convergence guarantee.
> > >
> > > Anyway, I appreciate your feedback and will consider that in the next discussion stage.
> > >
> > > [1] Bonawitz, Keith, et al. "Practical secure aggregation for privacy-preserving machine learning." proceedings of the 2017 ACM SIGSAC Conference on Computer and Communications Security. 2017.
> > >
> > > [2] Bell, James Henry, et al. "Secure single-server aggregation with (poly) logarithmic overhead." Proceedings of the 2020 ACM SIGSAC Conference on Computer and Communications Security. 2020.
> > >
> > > [3] Yaldiz, Duygu Nur, Tuo Zhang, and Salman Avestimehr. "Secure Federated Learning against Model Poisoning Attacks via Client Filtering." ICLR 2023 Workshop on Backdoor Attacks and Defenses in Machine Learning.

---

> > > > ### Author Response · Authors · 2024-08-14
> > > >
> > > > Thank you for your additional comments.
> > > >
> > > > We acknowledge that SecAgg (Google's original work), which uses pairing masking with secret sharing technologies, can address client-server collusion. However, this is just one research direction in the field of secure aggregation within FL. Other approaches such as HE, functional encryption, additive secret sharing, and conventional MPC are also significant and valuable contributions to this field. As we have illustrated above, these methods remain practical. We want to emphasize that not all non-SecAgg works are impractical or unacceptable. We kindly ask the reviewer to consider this point.
> > > >
> > > > The cosine scores do not necessarily improve with higher values. In our released open-source code, we have selected the threshold using an adaptive approach.
> > > >
> > > > Regarding convergence proof, we agree with the reviewer that our similarity-based approach differs slightly from ShildFL. Due to limited response time, we may not provide a full theoretical proof at this moment; however, our experiments on three datasets demonstrate experimental convergence guarantees. A detailed proof will be included in the final version.
> > > >
> > > > Concerning DP concerns raised by another reviewer: Our experimental results show it does not affect benign client selection.
> > > >
> > > > Finally, thank you again for your further comments—they have indeed helped us improve the quality of our paper. We hope our responses address these concerns and positively influence the final decision in this round of discussions.

---

### Official Review · Reviewer_LqQw · 2024-07-17

**Soundness:** 3
**Presentation:** 3
**Contribution:** 2
**Rating:** 6
**Confidence:** 5

**Summary:**

This paper introduces Dual Defense Federated Learning (DDFed), a framework designed to tackle two major challenges in federated learning: privacy breaches and poisoning attacks. By integrating fully homomorphic encryption, DDFed securely aggregates model updates, thereby enhancing privacy protection without the need for impractical non-colluding two-server setups. Furthermore, it incorporates a two-phase anomaly detection mechanism for encrypted model updates, which includes secure similarity computation and feedback-driven collaborative selection.

**Strengths:**

In contrast to most works that study either privacy or security, this paper addresses both in FL simultanously. The paper is well-written and well-structured. The method is thoroughly evaluated through extensive experiments in various FL scenarios and poisoning attacks. The practical implication of not requiring non-colluding servers makes the approach more feasible for real-world applications.

**Weaknesses:**

The novelty of the work primarily lies in the combination of techniques rather than theoretic novelty in individual components. The multiple steps and interactions between clients and the server could introduce significant computational and communication overhead. This raises questions about the framework’s scalability in large-scale FL settings with thousands of clients. Furthermore, the paper does not provide a detailed comparison of the computational costs and latency introduced by FHE and the anomaly detection mechanism compared to baseline methods.

**Questions:**

1. What is the time complexity on server-side and client-side separately?
2. How does the proposed method scale with the number of clients?
3. What is the threshold for the proportion of compromised clients that DDFed can effectively handle?

**Limitations:**

Yes

---

> ### Author Response · Authors · 2024-08-06
>
> We appreciate the reviewer's positive decision and those raised concerns. The challenge of the paper lies in resolving the dilemma where detecting model poisoning requires plaintext model updates from each client, while privacy protection demands safeguarding these updates.
>
> This is our first attempt to address this dilemma using a homomorphic encryption scheme as the privacy-enhancing technology. Unlike other methods such as secret sharing, pairwise masking, and functional encryption, the native design of the homomorphic encryption-based secure aggregation approach prevents the aggregation server from obtaining both anomaly detection results and aggregated models. Thus, such a design results in introducing an additional interaction step compared to the original federated learning paradigm and incurs extra time costs due to privacy protection. However, this is inevitable. Existing privacy-preserving federated learning solutions typically sacrifice either computational efficiency or communication efficiency—or both—to provide strong privacy and security guarantees.
>
> In our paper, we note that feedback-driven collaborative selection requires an additional round of interaction due to the native security model of homomorphic encryption. In our next research plan, we aim to use advanced cryptographic computation technologies like functional encryption, which have different security models and can eliminate the need for extra interaction rounds.
>
> From a computational cost perspective, the use of homomorphic encryption for privacy enhancement does not add significant overhead actually. This is contrary to what most non-cryptography researchers might expect, as shown by the initial time cost results in Table 1.
> Apart from the inherent challenges of developing homomorphic encryption technology, we believe the minor computational overhead reported in Table 1 is due to the following factors:
> 1) The scale of homomorphic encryption used is relatively small. The total parameters for evaluating models on the MNIST and FMNIST datasets are about 0.23 million. As requested by other reviewers, we will also include evaluations on larger datasets and models such as CIFAR10 in the final version.
> 2) Our design mechanism does not involves complex computations over the encrypted model, with only one layer of multiplication depth. It is well-known that multiplication depth significantly impacts efficiency in homomorphic encryption.
> Besides, although we provided a computational cost comparison with other approaches, we did not measure latency. As shown in the released open-source code, we implemented a simulated FL framework rather than an actual FL system at this prototype stage. Our goal is to validate the effectiveness of the proposed dual defense approaches. Building an FL system in a distributed real-world network environment deviates from the core focus of this paper and will be addressed in future research stages.
>
> Here are specific responses to the raised questions:
> 1) Compared to native FL paradigms or existing non-private solutions, our work introduces homomorphic encryption to enhance privacy preservation and support poisoning model detection. Due to the introduction of homomorphic encryption, providing a precise formal time complexity analysis is challenging. We believe the reviewer would like a breakdown of the time costs involved. In addition to the original federated learning training time, extra time is required for homomorphic encryption processes such as encryption, decryption, and secure computation (i.e., inner-product and addition among encrypted models and fusion weights or perturbation noise). Each client is responsible only for encryption and decryption, while all secure computations are performed by the server.
> 2) Due to limited experimental conditions, we only report results for client sizes ranging from 10 to 100. Theoretically, our proposed framework supports a larger scale of clients. Since the input sizes for encryption, decryption, and secure computation are linear relative to the number of clients, the time cost is also linear based on current reported results. For example, in evaluating the MNIST dataset as shown in Table 1, compared to a non-private solution, our approach requires approximately an additional 2 seconds per training round with 100 clients participating in FL training. Therefore, scaling up to thousands of clients may take about 20 seconds per round without considering network latency.
> 3) As stated in Section 3.1 regarding threat assumptions (i.e., the threat model in security and privacy papers), our work can handle scenarios where less than half of the clients are compromised. We will further clarify these statements and analyses in the final version.
>
> We hope we have addressed the reviewer's concerns. We will improve the final version of this paper and thank the reviewers for their valuable comments.

---

> > ### Comment · Reviewer_LqQw · 2024-08-09
> >
> > Thanks for the clarification. Although my questions have been answered, my opinion regarding the contribution has not changed. Therefore, my decision remains unchanged.

---

> ### Author Response · Authors · 2024-08-10
>
> Still, thank you for the valuable comments given earlier.

---

### Decision · Program_Chairs · 2024-09-25

**Decision:**

Accept (poster)

**Comment:**

Thank you for your submission. The reviewers agree that the paper provides an interesting and novel idea. The problem definition is important, and the combination of FHE with differential privacy seems to be moderately practical, at least for smaller models (this can perhaps be justified because small models are the only kinds of models that can be trained in client-level federated learning deployments on mobile devices). At the same time, the reviewers pointed out some concerns. I believe that if the authors can address the first one in particular, the paper would be significantly strengthened.

(1) The evaluation seems to be limited in terms of the attacks and defenses evaluated. In the cat-and-mouse game of backdoor defenses, it is important (and common) to compare against a wide swath of both attacks and defenses, particularly in the absence of theoretical robustness guarantees. The authors have added an adaptive attack during the rebuttal, which is a good start, but there is a wide literature of attacks and I would like to see a more thorough review, and evaluation against relevant attacks. Additionally, they have not addressed the comment of reviewer zJWB to add additional defenses. In particular, I am concerned that this defense algorithm may not compare favorably to more recent defenses than the baselines evaluated in this work. For example, in addition to the methods listed by reviewer zJWB (which are indeed important reference points), how does this defense compare to methods such as [1-3]? (This is just a short list, please conduct a more thorough review). What do you lose in terms of backdoor robustness by adding your secure aggregation technique? This field is moving very fast, and a comprehensive evaluation requires a more thorough comparison to related work.
[1] Pillutla et al, Robust aggregation for federated learning
[2] Wang  et al, Towards a defense against federated backdoor attacks under continuous training
[3] Zhang et al, Flip: A provable defense framework for backdoor mitigation in federated learning

Note: even if DDFed performs worse than these other defenses in terms of backdoor robustness, it's not a deal-breaker in my view, because the scheme is providing other benefits, namely secure aggregation. However, I do think it's important to include this empirical evaluation for completeness.

(2) There were some concerns about the threat model, and the assumption that clients do not collude with the server. While I agree that this is difficult to ensure in practice, it is also true that many papers in the literature make this assumption. Hence, this concern did not factor into my evaluation.